

# Not all brawn, but some brain. Strength gains after training alters kinematic motor abundance in hopping

Bernard X.W. Liew[1], Andrew Morrison[2], Hiroaki Hobara[3], Susan Morris[4] and Kevin Netto[4]

[1] Centre of Precision Rehabilitation for Spinal Pain (CPR Spine), School of Sport, Exercise and Rehabilitation Sciences, College of Life and Environmental Sciences, University of Birmingham, Birmingham, United Kingdom

[2] Sport and Exercise Sciences, Faculty of Science and technology, Anglia Ruskin University, Cambridge, United Kingdom

[3] Human Informatics Research Institute, National Institute of Advanced Industrial Science and Technology (AIST), Tokyo, Japan

[4] School of Physiotherapy and Exercise Sciences, Curtin University of Technology, Perth, Western Australia, Australia

Corresponding author
Bernard X.W. Liew,
LiewB@adf.bham.ac.uk

## ABSTRACT

**Background**. The effects of resistance training on a muscle's neural, architectural, and mechanical properties are well established. However, whether resistance training can positively change the coordination of multiple motor elements in the control of a well-defined lower limb motor performance objective remains unclear. Such knowledge is critical given that resistance training is an essential and ubiquitous component in gait rehabilitation. This study aimed to investigate if strength gains of the ankle and knee extensors after resistance training increases kinematic motor abundance in hopping.

**Methods**. The data presented in this study represents the pooled group results of a sub-study from a larger project investigating the effects of resistance training on load carriage running energetics. Thirty healthy adults performed self-paced unilateral hopping, and strength testing before and after six weeks of lower limb resistance training. Motion capture was used to derive the elemental variables of planar segment angles of the foot, shank, thigh, and pelvis, and the performance variable of leg length. Uncontrolled manifold analysis (UCM) was used to provide an index of motor abundance (IMA) in the synergistic coordination of segment angles in the stabilization of leg length. Bayesian Functional Data Analysis was used for statistical inference, with a non-zero crossing of the 95% Credible Interval (CrI) used as a test of significance.

**Results**. Depending on the phase of hop stance, there were significant main effects of ankle and knee strength on IMA, and a significant ankle by knee interaction effect. For example at 10% hop stance, a 1 Nm/kg increase in ankle extensor strength increased IMA by 0.37 (95% CrI [0.14–0.59]), a 1 Nm/kg increase in knee extensor strength decreased IMA by 0.29 (95% CrI [0.08–0.51]), but increased the effect of ankle strength on IMA by 0.71 (95% CrI [0.10–1.33]). At 55% hop stance, a 1 Nm/kg increase in knee extensor strength increase IMA by 0.24 (95% CrI [0.001–0.48]), but reduced the effect of ankle strength on IMA by 0.71 (95% CrI [0.13–1.32]).

**Discussion**. Resistance training not only improves strength, but also the structure of coordination in the control of a well-defined motor objective. The role of resistance

training on motor abundance in gait should be investigated in patient cohorts, other gait patterns, and its translation into functional improvements.

## INTRODUCTION

Regular participation in walking and running has important health benefits (*Lee et al., 2014*), and is commonly undertaken along irregular surfaces. Normally, humans have no problems maintaining dynamic postural control and energy efficiency during gait despite these surface irregularities, ensuring a smooth center of mass (COM) trajectory. Excessive COM trajectory disturbance in gait can be energetically costly and potentially destabilizing to postural control (*Andrada et al., 2013*; *Geyer, Seyfarth & Blickhan, 2006*).

Perturbation to the COM trajectory can be minimized over irregular surfaces by adjusting the length of a simplified virtual leg (henceforth termed as leg), spanning the COM to the center of pressure (COP) (*Andrada et al., 2013*; *Geyer, Seyfarth & Blickhan, 2006*). Leg length is regulated by four major segments (foot, shank, thigh, and pelvis), along which flexion-extension occurs. The excess of segments required to control a single leg, means that the body has an abundance of solutions to flexibly combine segment angles to achieve the same leg length (*Auyang, Yen & Chang, 2009*). Greater motor abundance in leg length regulation affords the body greater adaptability to rapidly react to irregular surfaces to minimize COM trajectory perturbation. In the context of quantifying motor abundance in leg length regulation, the Uncontrolled Manifold (UCM) analysis has been used to investigate the motor control of unilateral hopping (*Auyang, Yen & Chang, 2009*). UCM provides a ratio of two variances: one where the variance in angles (motor elements) does not change leg length (performance variable)—Goal-Equivalent Variance (GEV), to a variance in angles which change leg length—Non Goal Equivalent Variance (NGEV) (*Auyang, Yen & Chang, 2009*).

The manifestation of normal abundance in motor task may depend on the task's physical demand relative to an individual's physiological strength capacity (*Greve et al., 2013*; *Olafsdottir, Zatsiorsky & Latash, 2008*; *Park, Han & Shim, 2015*; *Shim et al., 2008*; *Yen & Chang, 2010*). Greater motor abundance may emerge when the task's relative physical demand increases (*Greve et al., 2013*). For example, older adults with lower maximal strength have non-significantly greater motor abundance in sit-to-stand compared to younger adults with greater maximal strength (*Greve et al., 2013*). When an external load was added to walking, there was a significant increase in the motor abundance of joint angle co-variation in the control of the COM trajectory in the frontal plane, and a non-significant increase in abundance in the control of the COM trajectory in the sagittal plane (*Qu, 2012*). It is reasonable to expect that if one muscle is operating near its physiological limit, additional muscles would be recruited to achieve successful performance.

In contrast, prospective resistance training studies of the upper limb demonstrated that greater wrist and finger strength was associated with greater abundance in finger force

coordination tasks (*Olafsdottir, Zatsiorsky & Latash, 2008*; *Park, Han & Shim, 2015*; *Shim et al., 2008*). Resistance training may augment motor abundance via several mechanisms: (1) by increasing reciprocal inhibition of co-varying muscle groups via heteronomous spinal pathways (*Geertsen, Lundbye-Jensen & Nielsen, 2008*), (2) increasing the role of bi-articular muscles in inter-segmental kinematic co-variation (*Cleather, Southgate & Bull, 2015*), and (3) increasing the number of muscle modes available for co-variation (*Hashiguchi et al., 2016*).

Although prospective study designs (*Olafsdottir, Zatsiorsky & Latash, 2008*; *Park, Han & Shim, 2015*; *Shim et al., 2008*) already provide a higher level of evidence base than a cross-sectional design (*Greve et al., 2013*; *Qu, 2012*), the relationship between physiological strength and motor abundance may still have been confounded by other factors. First, different mathematical formulation of the variance ratios of motor abundance may have contributed to the conflicting evidence (*Greve et al., 2013*; *Olafsdottir, Zatsiorsky & Latash, 2008*). For example, *Olafsdottir, Zatsiorsky & Latash (2008)* defined motor abundance using the ratio of $\frac{GEV-NGEV}{GEV+NGEV}$ (with GEV and NGEV divided by their respective degrees of freedom), whilst *Greve et al. (2013)* used a simple ratio of $\frac{GEV}{NGEV}$. Second, differences may lie in the hierarchical level of analysis as it pertains to the neural control of motor abundance. *Greve et al. (2013)* investigated the covariation of joint-level kinematics to whole-body performance variables, such as ground reaction force. The analysis by *Greve et al. (2013)* thus did not consider an intermediary level of hierarchical control—that is in the stabilization of limb-level performance variables (*Toney & Chang, 2016*). In contrast, *Olafsdottir, Zatsiorsky & Latash (2008)* investigated the covariation of limb-level (finger) force in the stabilization of the total force generated by four fingers. To this end, a prospective study design that quantifies lower limb motor abundance within a hierarchical control framework would increase the evidence base behind using resistance training to improve motor abundance.

Resistance training is an essential component in gait rehabilitation (*Papa, Dong & Hassan, 2017*). Whether a gain in physiological strength capacity benefits or harms motor abundance is an essential question to answer, as it directly implicates the role of resistance training in gait rehabilitation. The aim of this study was to investigate if lower limb strength gains after resistance training influenced lower limb motor abundance. Hopping represents an excellent model of forward gait patterns to fulfill the present study's aim. First, lower limb spring-mass dynamics in hopping is present in walking and running (*Geyer, Seyfarth & Blickhan, 2006*). Second, leg length is a regulated performance variable in hopping using UCM analysis (*Auyang, Yen & Chang, 2009*). Given that leg length mechanics contributes to COM trajectory (*Moritz & Farley, 2003*), greater motor abundance in leg length stabilization in hopping may translate into more available motor solutions to minimize COM trajectory perturbation when walking or running over irregular surfaces. Third, normal inter-segmental kinematic and kinetic coordination in hopping, quantified using UCM and vector coding, varies depending on the task's relative physical demand (*Auyang, Yen & Chang, 2009*; *Smith, Popovich Jr & Kulig, 2014*; *Yen & Chang, 2010*). Similar to the effects of finger muscle strengthening on finger pressing motor abundance (*Olafsdottir, Zatsiorsky & Latash, 2008*), the hypothesis of this study was that

greater strength gains after resistance training would increase kinematic motor abundance in leg length regulation during hopping.

## MATERIALS AND METHODS

### Participants

The data presented in this manuscript represents the pooled group results of a sub-study from a larger project investigating the effects of resistance training on load carriage running energetics (*Liew, Morris & Netto, 2017*). Healthy adult recreational runners between 18 to 60 years old were invited to participate in the study. Participants had to be actively engaged in running or running-related sports with a minimum cumulated total duration of 45 min per week to be considered for inclusion. Exclusion criteria included: (1) self-reported medical conditions which precluded the safe performance of running, jumping, hopping activities and heavy resistance exercises; (2) self-reported running related injuries currently and within the past three months; (3) surgeries within the past year; and (4) females who were pregnant at time of recruitment. Thirty participants volunteered for this study (16 male, 14 female). This study was approved by the Curtin University Human Research Ethics Committee (RD-41-14). Informed written consent was sought and gained prior to study enrolment.

### Intervention

The two training programs were developed to improve load carriage running energetics (Table S1). One group performed "conventional" heavy-resistance isoinertial training on the bilateral leg press, unilateral calf raises, and lunge exercises. These exercises have been routinely adopted in conventional load-carriage military training (*Knapik et al., 2012*). The other group performed "load carriage specific" resistance training targeting the specific biomechanical requirements of load carriage running (*Liew, Morris & Netto, 2017*). Exercises in this group comprised of externally loaded single-leg hopping to increase leg stiffness, countermovement jumps to increase knee power generation, and hip flexor pull to increase pre-swing running energetics (*Liew, Morris & Netto, 2017*). Greater leg stiffness, knee power generation, and pre-swing hip energetics were previously shown to be required to sustain constant running velocity during load carriage (*Liew, Morris & Netto, 2016a*; *Silder, Besier & Delp, 2015*). Despite the differences between the training programs, the present study was only interested in accounting for the between time (pre-post) change in strength (ankle extensor = mean increase 0.34 Nm/kg (95% CI [0.25–0.42] Nm/kg); knee extensor = mean increase 0.24 Nm/kg (95% CI [0.11–0.37] Nm/kg) (*Liew, Morris & Netto, 2017*), in predicting alterations in hopping motor abundance.

### Three dimension motion capture on hopping (combined group analysis)

Participants performed unilateral hopping, on both sides, at a self-selected frequency lasting approximately 15 s. During hopping, the arms were held in a 90° abducted position, to allow visualization of the lateral pelvic markers. The only instruction provided was to hop at a "comfortable pace". In the post hoc analysis stage, only hops maintained within 10%

of the individual's mean hop frequency was kept for further analysis (termed as successful trials). This 10% frequency window was deemed appropriate given that a previous study reported a variation of up to 20% for adults hopping at their preferred frequency (*Beerse & Wu, 2016*). A between side standing rest period of one minute was provided. An 18 camera motion capture system (Vicon T-series, Oxford Metrics, UK) (250 Hz), with synchronized in-ground force plates (AMTI, Watertown, MA, USA) (2000 Hz) were used to collect marker trajectories and force data (Vicon Nexus, v2.3, Oxford Metrics, UK). Force data were used to detect initial contact and toe-off, with a 20 N vertical force threshold used. The marker placements were based on a previous study (*Liew et al., 2016b*). A seven-segment lower-limb biomechanical model was created in Visual 3D (C-motion, Germantown, MD, USA) (*Liew, Morris & Netto, 2016a*). Joint centers of the hip were derived using a regression equation (*Bell, Brand & Pedersen, 1989*), whilst those of the knee and ankle were derived as the midpoint between the medial and lateral femoral condyles, and malleoli, respectively. Segment inertial and geometric properties were based on Visual 3D's default routines. The biomechanical model's position and orientation was derived using inverse kinematics. Each joint had three rotational degrees of freedom, with the model having a total of 18 degrees of freedom. The laboratory and joint coordinate system used had the following sequence: $X$ axis—mediolateral with positive pointing to the right, $Y$ axis—postero-anterior with positive pointing anteriorly, and $Z$ axis—vertical with positive pointing proximally. Marker trajectories were low pass filtered at 12 Hz (zero lag, 4th order, Butterworth).

## Isokinetic strength measurement

Isokinetic concentric strength testing of the bilateral knee and ankle extensors was performed in a dynamometer (HUMAC NORM; Computer Sports Medicine Inc., Stoughton, MA, USA), collecting data at 100 Hz and set up according to the manufacturer's guidelines (*Liew, Morris & Netto, 2017*). For each muscle group tested, participants first performed 10 repetitions of warm-up contractions at 90°/s, and two sets of six maximal concentric-concentric contractions at 60°/s. Each set was interspersed with one minute of seated rest in-situ. Between muscle group and side rest periods of three minutes were provided.

## Uncontrolled manifold analysis

A modified sagittal plane forward kinematic model mapping segment angles to leg length used in a previous study was adopted in the present study (*Auyang, Yen & Chang, 2009*) (Fig. 1). Leg length was presently defined by the vector between the centre of pressure (COP) to the proximal end of the pelvic segment, instead of the toe and anterior superior iliac spine markers, respectively (*Auyang, Yen & Chang, 2009*). The X-coordinate of landmarks used to create the planar segments was set to zero. The foot, shank, thigh, and pelvic planar segments were defined by the line vectors in the YZ plane between (1) COP to ankle joint centre, (2) ankle to knee joint centre, (3) knee to hip joint centre, and (4) hip joint to proximal end of the pelvic inertial segment. Trial-to-trial variability in leg length can be influenced by variable changes to segment lengths, given the presence of soft tissue artefact. This effect was minimized by using landmarks modeled after the biomechanical model was

 

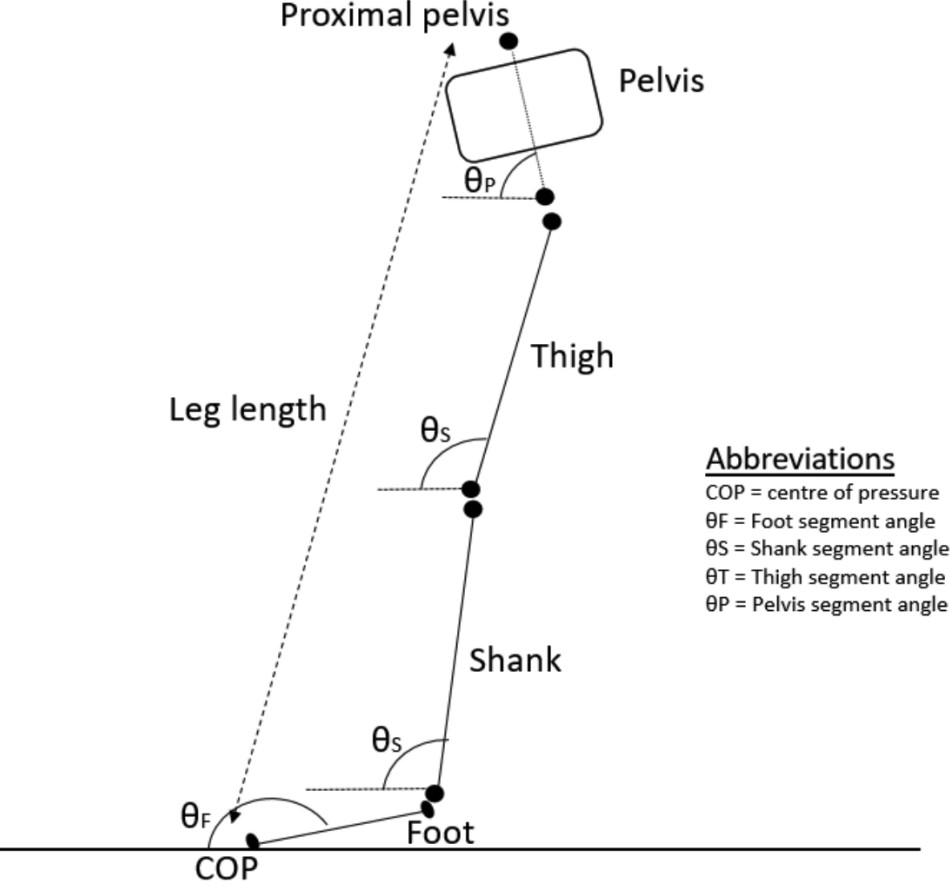

**Figure 1  Planar kinematic model.**

optimized using inverse kinematic. Sagittal planar angles of each segment in the YZ plane were defined relative to the laboratory's horizontal plane, using the Right Hand Rule. All segment planar angles and leg length were time-normalized to 100 data points in the stance period for UCM analysis.

The UCM analysis was carried out using a previously published method (*Auyang, Yen & Chang, 2009*), for each of the 100 stance data points.

$$NGEV = \frac{trace\left(orth\left(J^t\right)^t.C.orth\left(J^t\right)\right)}{d}\dots \tag{1}$$

$$GEV = \frac{trace\left(null\left(J\right)^t.C.null\left(J\right)\right)}{n-d}\dots \tag{2}$$

$$IMA = \frac{GEV-NGEV}{GEV+NGEV}\dots \tag{3}$$

In Eqs. (1) to (3), non-goal equivalent variance (NGEV) represented the variance of all segment angle combinations that contributed to leg length changes while goal equivalent variance (GEV) represented the variance of all segment angle combinations that did not change leg length. The index of motor abundance (IMA) represented the ratio of

two variance measures (*Auyang, Yen & Chang, 2009*). Thus an IMA > 0 characterized variation in segment angles which minimized leg length variation (i.e., motor abundance), and an IMA < 0 characterized variation in segment angles that maximized leg length variation (*Auyang, Yen & Chang, 2009*). In the equation, J is the Jacobian matrix mapping infinitesimally small changes in segment angles to changes in leg length; C is the co-variance matrix in the deviation of the segment angles from the mean reference segment angles at each datum; d is the degree of freedom in the performance variable ($d = 1$ in this study); and n is the degree of freedom in the elemental variables ($n = 4$).

## Statistical analysis (combined group inference)

A previous study reported that the standard deviation of the GEV and NGEV was lowest with at least 20 trials (*Latash et al., 2010*). Hence, UCM analysis and subsequent functional regression analyses was performed only on participants with ≥20 successful trials. Simple linear regression was used to quantify differences in age, height, weight, running frequency (times/week) and cumulated distance (km/week) over the past six weeks, baseline ankle and knee extensor strength between participants with and without ≥20 successful trials.

Descriptive scalar variables of post-pre change in hopping frequency and stance duration, and baseline (pre-training) waveform variables of leg length, foot, shank, thigh, and pelvic segment angles, IMA, GEV, and NGEV were reported for participants with ≥20 successful hopping trials. The dependent variable was the between time change in waveform IMA. The predictor variables were the between time change (post-pre) in ankle, knee strength and their interaction, and the regression coefficients were adjusted for three covariates: (1) change (post-pre) in hopping frequency, (2) side (right vs. left); and (3) total number (post + pre) of hopping trials were included in the statistical model. These statistical adjustments were made given that changes in IMA between pre- and post-testing could be due to (1) changes in hopping frequency, (2) limb dominance, and (3) the number of trials used for UCM analysis; and we wish to isolate the estimate of strength gains on IMA changes. Bayesian regression functional analysis was performed in R software (*Goldsmith & Kitago, 2016*). Recent investigations in sports science have advocated the avoidance of frequentist null-hypothesis significance testing, and instead to focus on estimating the probabilities associated with observing an effect size. Fixed effect parameters for ankle and knee strength, frequency, side, trial number, and non-parametric smooth functions (modelled with 15 B-splines) were estimated using a Gibbs sampler with a burn-in of 1,000 and drawing 15,000 inference samples. The residual covariance structure was estimated using Bayesian functional principle components. A significant effect was defined by a non-zero crossing of the Bayesian 95% credible interval (CrI).

## RESULTS

Twenty-five participants had ≥20 successful trials (Fig. 2). No significant differences in baseline characteristics between participants with and without ≥20 successful trials, were detected (Table 1). For the 25 participants with ≥20 successful trials, the number of hop trials used for UCM analysis per participant ranged from 20 to 57.

![PeerJ]

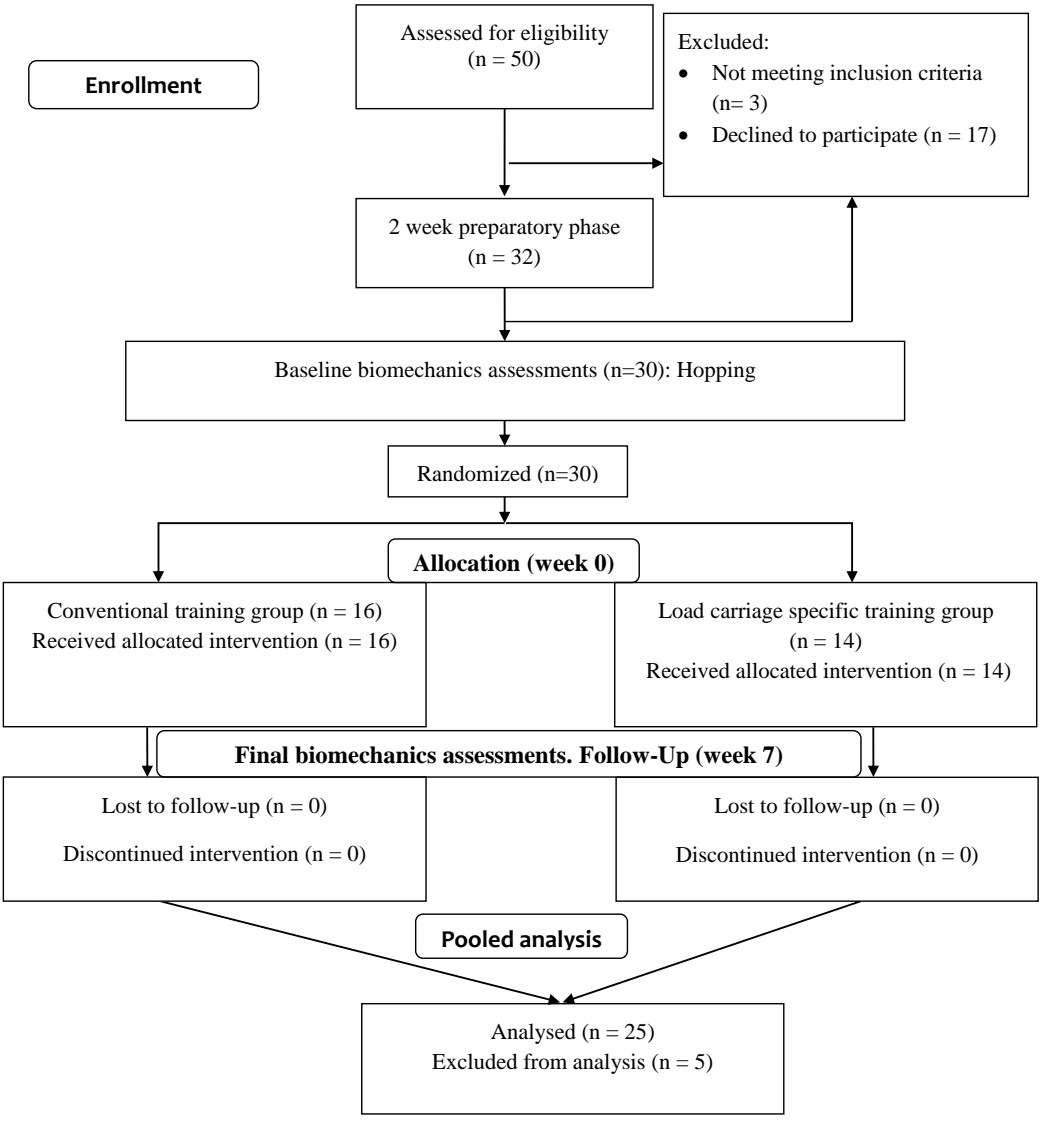

**Figure 2** CONSORT Diagram.

Baseline hopping kinematics, IMA, GEV, NGEV are reported in Figs. 3 and 4. For the 25 participants with ≥20 successful trials, the mean (standard deviation) change in hopping frequency was a 0.15 (0.25) Hz increase, and change in stance duration was a 0.01 (0.03) s decrease post-testing, relative to a baseline of 2.24 (0.26) Hz and 0.31 (0.04) s, respectively.

For simplicity, only effects at a discrete hop phase within a statistically significant temporal period are reported here. At 10% hop stance, there was a significant interaction between ankle and knee strength gains, and significant main effect of ankle and knee strength gains (Fig. 5). A 1 Nm/kg increase in ankle extensor strength increased IMA by 0.37 (95% CrI [0.14–0.59]), a 1 Nm/kg increase in knee extensor strength decreased IMA by 0.29 (95% CrI [0.08–0.51]), but increased the effect of ankle strength on IMA by 0.71 (95% CrI

**Table 1  Baseline characteristics of participants.**

|  | ≥20 hop trials for UCM (n = 25) | <20 hop trials for UCM (n = 5) | p value |
|---|---|---|---|
| Age (years) | 30.5 (9.7) | 28.6 (6.5) | 0.684 |
| Body mass (kg) | 67.1 (12.2) | 74.9 (11.1) | 0.196 |
| Height (cm) | 171.3 (7.6) | 176.1 (7.5) | 0.210 |
| Running frequency over past 6 weeks (times/week) | 2.7 (1.4) | 2.2 (1.3) | 0.423 |
| Running distance over past 6 weeks (km/week) | 16.8 (18.8) | 15.2 (10.1) | 0.852 |
| Ankle strength (Nm/kg) | 1.07 (0.19) | 1.07 (0.24) | 0.978 |
| Knee strength (Nm/kg) | 2.05 (0.39) | 2.09 (0.42) | 0.786 |

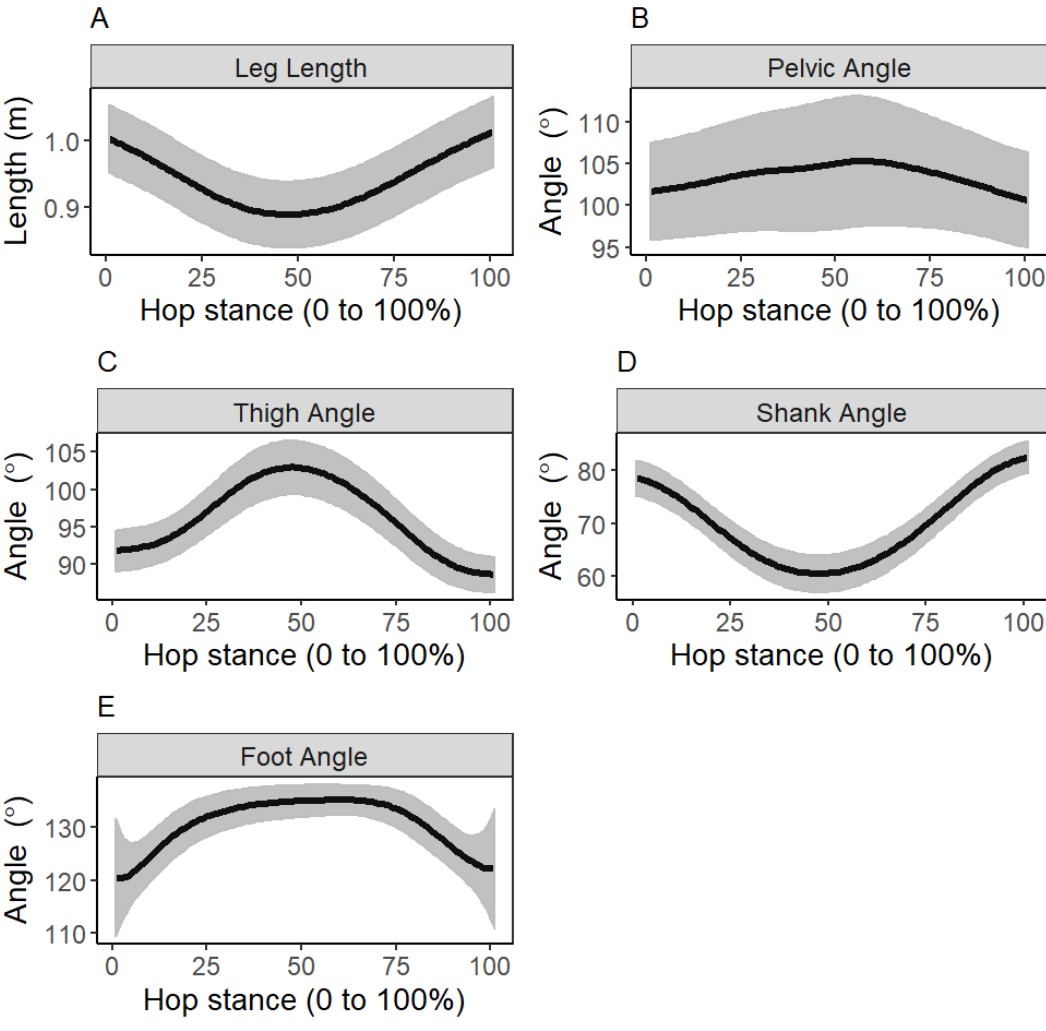

**Figure 3  Baseline mean (standard deviation as error clouds) of leg length and segment angles.** (A) Leg length, (B) pelvic angle, (C) thigh angle, (D) shank angle, (E) foot angle.

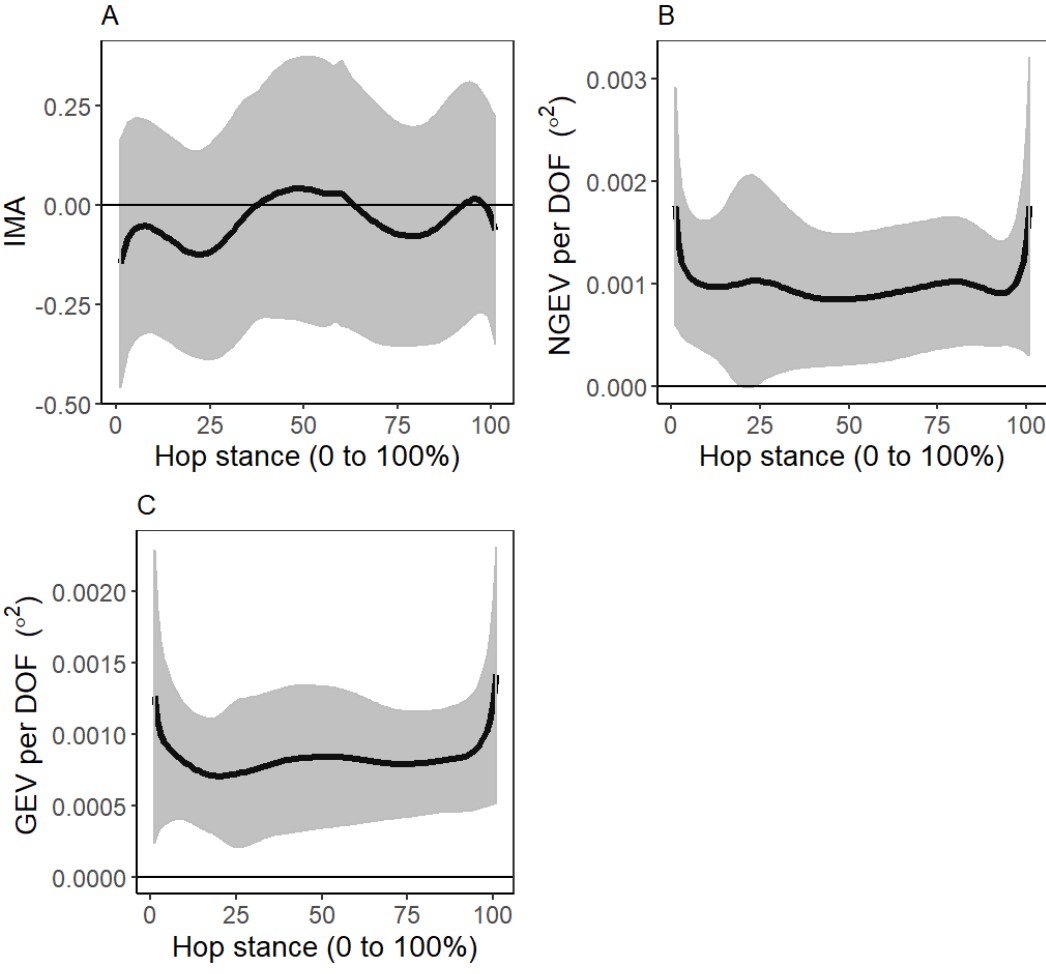

**Figure 4** Baseline mean (standard deviation as error clouds) of (A) Index of Motor Abundance (IMA), (B) Non-Goal Equivalent Variance (NGEV) and (C) Goal Equivalent Variance (GEV).

[0.10–1.33]) (Fig. 5). At 55% hop stance, a 1 Nm/kg increase in knee extensor strength increase IMA by 0.24 (95% CrI [0.001–0.48), but reduced the effect of ankle strength on IMA by 0.71 (95% CrI [0.13–1.32]) (Fig. 5). At 70% hop stance, a 1 Nm/kg increase in ankle extensor strength reduced IMA by 0.31 (95% CrI [0.06–0.58]) (Fig. 5). At 98% hop stance, a 1 Nm/kg increase in ankle extensor strength increased IMA by 0.39 (95% CrI [0.05–0.73]) (Fig. 5).

## DISCUSSION

Leg length regulation is a strategy of coping with irregular surfaces to minimize disturbance to the COM trajectory during gait (*Andrada et al., 2013*; *Geyer, Seyfarth & Blickhan, 2006*). Normally, leg length regulation is achieved by harnessing segmental kinematic motor abundance (*Auyang, Yen & Chang, 2009*). Even though resistance training has been typically prescribed to treat gait impairments, there is uncertainty as to the

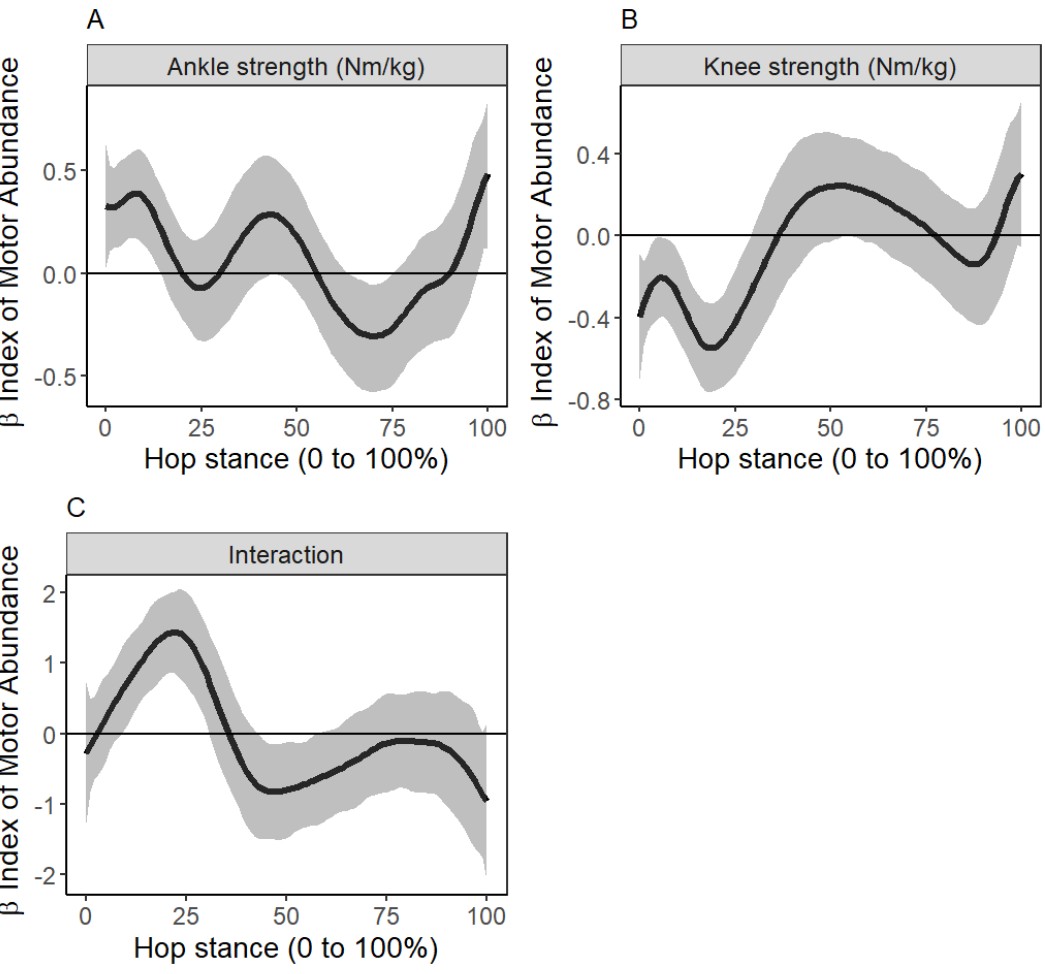

**Figure 5** Mean (95% Credible Interval as error clouds) of beta coefficient of 1 Nm/kg increase in ankle extensor strength (A) and knee extensor strength (B), and its interaction (C).

relationship between physiological strength capacity and normal motor abundance. This poses a dilemma as to whether resistance training benefits or harms gait motor control rehabilitation. In this study, we prospectively investigated if lower limb strength gains after resistance training, predicted a change in IMA during a simple model of spring-mass gait—unilateral hopping. In partial agreement with our hypothesis, greater strength gains predicted an increase in IMA, but this effect was dependent on the muscles being strengthened. In addition, the effects of ankle strength gained on IMA was opposite to that of knee strength gains.

The increase in kinematic motor abundance with an isolated gain in ankle extensor strength after initial contact and toe-off of hopping, was consistent with the findings of previous resistance training studies of the upper limb (*Olafsdottir, Zatsiorsky & Latash, 2008*; *Park, Han & Shim, 2015*; *Shim et al., 2008*). The mechanisms linking strength gains to motor abundance improvements was not investigated in the present study, but may speculatively involve mechanical and neural factors. The bi-articular gastrocnemius is

mechanically capable of plantar flexing the foot while flexing the knee (*Cleather, Southgate & Bull, 2015*). Greater gastrocnemius strength could augment inter-segmental kinematic co-variation, increasing motor abundance during hopping. Resistance training has also been shown to increase reciprocal inhibition of antagonistic muscles within a joint muscle pair (*Geertsen, Lundbye-Jensen & Nielsen, 2008*). However, no studies to the authors' knowledge have directly investigated the influence of resistance training on heteronomous reflex pathways, which would enable inter-muscular co-variation and facilitate hopping kinematic motor abundance.

It was previously suggested that greater motor abundance will emerge in tasks with greater relative physical demand (*Greve, Hortobagyi & Bongers, 2017*; *Greve et al., 2013*), which contradicts the present findings. It may be that the relationship between physical demand and motor abundance is non-linear with potentially plateauing effects. Both physiological weakening and strengthening may augment motor abundance depending in part on the task's absolute demand on the participants, and the capacity to use available motor elements. *Wang, Watanabe & Asaka (2017)* reported that older adults have preserved muscle motor abundance but were more delayed at their recruitment during rapid balance recovery, than younger adults. The tasks used by Greve and Colleagues were either much slower, or required a combination of low muscular force and fast movement speed (*Greve, Hortobagyi & Bongers, 2017*; *Greve et al., 2013*). It is plausible that the speed-force demands in previous studies were low (*Greve, Hortobagyi & Bongers, 2017*; *Greve et al., 2013*), such that muscle groups had sufficient time to stabilize the performance variable(s). In addition, if a muscle is operating at its physiological limit, it can no longer compensate for the reduction in activation of other muscles. In this instance, co-variation may still occur but only between muscles with adequate physiological strength reserve. The relationship between task demand and motor abundance may be better understood by investigating abundance at the level of muscle activations, including UCM analysis in a reduced subset of motor elements (*Toney & Chang, 2016*).

The period surrounding mid-stance in hopping is critical of leg length regulation for peak muscular force minimization (*Auyang, Yen & Chang, 2009*), which minimizes the energy expenditure and joint contact loads during hopping. If knee strength gain occurred in isolation, an increase in motor abundance at 55% stance was observed. However, if strength gains occurred at the ankle and knee, due to the statistical interaction, motor abundance was reduced. The detrimental effect of additional ankle strength gain could be due to the foot segment angle around mid-stance being nearly invariant (Fig. 3) (*Joao et al., 2014*). If the foot functions as a punctum fixum around mid-stance (*Joao et al., 2014*), the gastrocnemius is only able to flex and extend the thigh segment, without compensatory foot kinematics to stabilize overall leg length. The invariant foot-segment angle may instead shift the joint-level mediator of leg length stabilization to the knee during the period of mid-stance.

Greater knee extensor strength reduced motor abundance after initial contact, but augmented the incremental effect ankle strength gain on abundance for leg length control. It may be that the influence of knee extensor strength gain on motor abundance was shifted to the control of leg orientation (angle between the leg and ground) (*Auyang, Yen & Chang,*

*2009*). Leg orientation at initial contact may be critical as it determines the overall position of the force application relative to the COM in stance.

The reduction in kinematic motor abundance predicted by an increase in knee extensor strength after initial contact differed from a study investigating walking in individuals with and without knee osteoarthritis (OA) (*Tawy, Rowe & Biant, 2018*). Several reasons could account for the disagreement. *Tawy, Rowe & Biant (2018)* did not directly quantify the relationship between knee extensor strength and motor abundance. The occurrence of knee OA is associated with a range of neuromuscular deficits (*Mills et al., 2013*), and the importance of knee extensor strength to motor abundance cannot be ascertained from a between-groups comparison. Second, *Tawy, Rowe & Biant (2018)* used COM trajectory, while the present study used leg length, as the performance variable for UCM analysis. It must be emphasized that using both the COM and leg length as performance variables are equally valid. The organization of motor control may involve a hierarchical structure (*Latash, 2010*), where segment-level variation serve to stabilize limb-level outcomes, and inter-limb variation stabilizes whole-body outcomes. Thus, the present study focused only on limb-level motor control, while *Tawy, Rowe & Biant (2018)* performed UCM analysis across two layers of hierarchical control.

Several aspects of the present study's methodology need to be discussed in lieu of differences in reported IMA of the present study, with that of a previous work (*Auyang, Yen & Chang, 2009*). First, the number of hop cycles included in the present study was much lower than the 170 cycled used in *Auyang, Yen & Chang (2009)*. This may explain the difference in IMA values between studies. Second, leg length was defined starting from the COP in the present study, but from the toe marker in *Auyang, Yen & Chang (2009)*. COP accuracy may be reduced when the magnitude of the GRF is small, which could explain the differences in IMA between the present study and *Auyang, Yen & Chang (2009)* during the periods surrounding initial contact and toe-off. However, the effective leg length during human locomotion may be more accurately defined from the point of ground force application, compared to the fixed toe-marker (*Coleman et al., 2012*). Despite this difference in leg length definition, the overall shape of the IMA reported in this study was similar to *Auyang, Yen & Chang (2009)*.

Previous studies provided evidence for the benefit of resistance training on finger force motor abundance (*Olafsdottir, Zatsiorsky & Latash, 2008*; *Park, Han & Shim, 2015*; *Shim et al., 2008*), and the results of the present study extends the evidence for the same benefit to the lower limb. Findings from the present study carry an optimistic message that strength training may benefit the rehabilitation of gait where movement coordination and strength are impacted upon by the presence of disease (*Hashiguchi et al., 2016*). By increasing motor abundance to stabilize leg length in hopping, resistance training may increase the adaptability of forward gait patterns over irregular surfaces. It is likely that different gait patterns require different joint-level and limb-level strengthening to benefit kinematic motor abundance, and this should be investigated in future studies. The present study's findings also demonstrate that local strength changes can influence movement coordination across the kinematic chain. Speculatively, this may imply that where strength gains cannot be feasibly achieved using a more functional form of strength training, a more

regionally focused form of training (e.g., open-kinetic chain exercises) can still have global functional benefits. Whether different strength training modes differentially influence lower limb motor abundance, remains to be investigated.

A limitation of this study was that the analysis predicting motor abundance from alterations in strength gains were analyzed using a prospective, pre-post design. However, we reduced the confounding factor of repeated measurement, by only including the effects of strength changes into the statistical model. A second limitation of this study was that the influence of strength gains on kinematic abundance was analyzed in healthy individuals. This limitation may in fact be a strength, as we were able to isolate the investigation of IMA changes to strength changes.

## CONCLUSIONS

In addition to the well-known effects on a muscle's neural, architectural, and mechanical properties, resistance training also influences the coordination of multiple motor elements in the control of a well-defined motor performance objective. The benefits of strength gain on motor abundance was dependent on the site of muscle strengthened and the phase of gait. The role of resistance training on motor abundance should be investigated in patient cohorts, other gait patterns, as well as its translation into functional improvements.

## ACKNOWLEDGEMENTS

The authors of this study would like to thank Nour Faiz Aqil Yaccob, Jason Hu, Nicholas Callaghan, Tess Moynihan, Hannah Watt, and Giorgia Alford for delivering the interventions. The results of this study are presented clearly, honestly, and without fabrication, falsification, or inappropriate data manipulation.

### Funding
The authors received no funding for this work.

### Competing Interests
The authors declare there are no competing interests.

### Author Contributions
- Bernard X.W. Liew conceived and designed the experiments, performed the experiments, analyzed the data, prepared figures and/or tables, authored or reviewed drafts of the paper, approved the final draft.
- Andrew Morrison analyzed the data, prepared figures and/or tables, authored or reviewed drafts of the paper, approved the final draft.
- Hiroaki Hobara analyzed the data, authored or reviewed drafts of the paper, approved the final draft.
- Susan Morris and Kevin Netto conceived and designed the experiments, authored or reviewed drafts of the paper, approved the final draft.

## Clinical Trial Ethics

The following information was supplied relating to ethical approvals (i.e., approving body and any reference numbers):

This study was approved by the Curtin University Human Research Ethics Committee (RD-41-14).

## Data Availability

The raw data and code are provided as Supplemental Files.

## Clinical Trial Registration

The following information was supplied regarding Clinical Trial registration:

ACTRN12616000023459

## Supplemental Information

Supplemental information for this article can be found online at http://dx.doi.org/10.7717/peerj.6010#supplemental-information.

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
