# Peer review of "Not all brawn, but some brain. Strength gains after training alters kinematic motor abundance in hopping"

_PeerJ, doi:10.7717/peerj.6010_

## Round 0.1 · original submission · Minor Revisions

Though one of the reviewers recommended major revisions, that review did not provide sufficient constructive feedback to constitute major changes in the manuscript. Nevertheless, the points raised are critical and should be addressed before final acceptance.

Also, please pay close attention to the detailed comments of reviewer one, whose in depth assessment can be used to improve clarity and exposition.

·

Basic reporting

I commend the authors on a manuscript which is very interesting, well written and novel. The raw data supplied with the manuscript allows readers to use or view the authors code to further understand the methods used in this study. This, together with the clear language used throughout the manuscript aids the reader to understand the complex uncontrolled manifold method. For this reason, I believe the study is appropriate for PeerJ and will be of interest its readers.

The abstract for this manuscript was clear and summarised the study well. The abstract also conforms to the journal standards. Enough background was given in the manuscript to allow the reader to understand why the research was undertaken, and what importance this research could have on the field. The literature referenced was relevant and recent and the formatting and language used throughout the manuscript was appropriate.

In general the figures were relevant to the work and well described. However, in Figures 2-4 I could not see any of the mean data points on the graphs, as the shaded ±SD of the data seemed to overlay this data. This must be amended in the revised version. I would also suggest the axes be amended in Figure 3, as explained in the general comments.

Experimental design

This study was well designed and appropriate methods were used to address the aims and objectives stated. The study is also original and provides new information to the field of motor control. Although the materials and methods were well written, I would suggest the clarification of a couple of points (as outlined in the general comments) to ensure that the methods are fully repeatable.

The study was approved by a local REC and informed consent was obtained from all participants.

Validity of the findings

The data reported in this study appears to be robust and statistically sound. My main concern is that the inclusion and exclusion criteria were not adequately described in this study, and should be clarified in the revised manuscript. In general however, the data was discussed well and the conclusions drawn were fair. Readers may benefit from a description on the potential implications this study may have in the discussion.

Additional comments

Main Points:
1. Clarification of the protocols used in recruitment and during data collection recommended.
2. It is not clear whether the excluded data led to statistical differences in the results as the authors have only described the differences as ‘small’.
3. It is not clear why the authors have used increases of 1Nm/kg as the benchmark for comparing to changes in IMA.
4. The authors have not argued their reason for analysing variable numbers of hop cycles between individuals (cited as ranging between 20 and 57 cycles).

Abstract
Line 18: I would suggest changing the word ‘manuscript’ in this sentence to ‘study’.

Introduction
Lines 64-68: Please ensure this sentence is all in the same tense (preferably past tense).
Line 73: Remove semicolon after reference
Line 75: Remove semicolon after reference
Line 79: Remove comma before ‘…may’
Lines 82-83: Please elaborate on the differences between these two papers. Some of the readers may not have access to these papers but may wish to understand the differences in order to understand the reasoning behind the methods used in your study.

Materials and Methods
Line 110: This section could be titled ‘Participants’.
Line 113: What are your definition of ‘healthy’ and ‘active’ here? How how can you be confident that all involved were active to similar extents? Was a thorough check of the individuals’ medical history taken to ensure that none have had any lower limb injuries which may affect their hopping biomechanics? Different levels of health/activity and baseline muscle strengths within the group could affect the interpretation of the mean results.
Line 113: Please clarify what the inclusion and exclusion criteria for the study were. This will aid in clarifying my comment above too.
Line 132: Please clarify here that your analysis was based on the combined group, and that you did not analyse both groups separately or chose to assess the data from one group only.
Line 144: Here, you have referenced another paper in which the biomechanical model is described. To ensure that the readers could replicate your methods from this paper alone, please mention the model used and describe whether this model adheres to the International Society of Biomechanics standards.
Furthermore, on inspection of the referenced paper, I find that it does not contain the marker placement protocol, as stated in this manuscript. Please insert the appropriate reference (Liew et al., 2015).
FYI:
Wu, G. & Cavanagh, P.R (1995). “ISB Recommendations in the Reporting for Standardization of Kinematic Data.” J Biomechanics 28(10) 1257–1261.
Grood, E.S., Suntay, W (1983). “A Joint Coordinate System for the Clinical Description of Three-Dimensional Motions. Application to the Knee” Transactions of the American Society of Mechanical Engineers 105:136–144.

Line 160: Please clarify what you mean by the ‘proximal end of the pelvic segment’. Is this calculated from the positions of both anterior superior iliac spines?
Line 170: The start of this reads as if a part of the sentence is missing. Although I understand that you are referring to the variables in the equations above, please refer to these equations in the text and consider capitalising the start of the sentence, or rephrasing slightly.
For example: ‘In the above equation (1), non-goal equivalent variance (NGEV) represents…’
Lines 189-191: Please insert appropriate reference here.

Results:
Line 199: Remove one of the words ‘baseline’ in this sentence.
Lines 200-204: Here, it is stated that the differences between the demographics and ankle/knee beta coefficients in the full group (n = 30) and the analysed group (n = 25) were ‘small’. Small differences can present statistical or clinical differences, which would influence the interpretation of the results. Please consider statistically comparing these variables to confirm that analyses carried out was valid.
Line 204: I recommend starting a new paragraph at ‘Baseline hopping kinematics…’ as these results refer only to the group analysed, unlike the previous sentences.
Lines 209-217: It is not clear why the authors have used increases of 1Nm/kg as the benchmark for comparing to changes in IMA. Was this arbitrarily chosen, or is this deemed to be the minimal clinically important change in strength?

Discussion:
Line 261: Missing word ‘gain’ after ‘…knee extensor strength…’
Line 289: This is the first place it is stated that the number of hop cycles analysed per participant varied between 20 and 57 cycles. Please also declare this in the results section.
More importantly, it would benefit the reader to understand why you chose this method of assessing the groups, rather than choosing x number of hop cycles (e.g. 20) to assess per individual. Could the method used not influence the data analysed and thus the interpretations made?

Figure 2:
Please add ± to legend in front of ‘standard deviation’ and add n number.
Please amend the figures so that the mean data points can be visualised.

Figure 3:
Please add ± to legend in front of ‘standard deviation’ and add n number.
Please amend the figures so that the mean data points can be visualised.
Please annotate the y axes of the individual graphs independently, so that it is perfectly clear which graph shows the ratio and which show the variances.

Figure 4:
Please amend the figures so that the mean data points can be visualised.

Other:
1. Consider moving Figure S1 into the main manuscript, as this is a good and informative figure.
2. Please add in a sentence or two in the discussion on the potential implications this study may have.
3. Often in this manuscript, a reference is cited in the middle of a sentence, where it could be placed at the end. Consider moving some of these references to the end of sentences so that they are easier to read. I realise this is an extremely minor comment (nit-picking) and is most definitely personal preference.

Reviewer 2 ·

Basic reporting

The authors analyzed motor abundance using segmental angles, not joint angles. Using segmental angles (rather than joint angles) is potentially a serious drawback because segmental angles cannot be changed independently: Changing one of the angles forces changes in other angles to preserve body integrity. This means that, even if there is no specific neural control strategy, co-variation among joint angles may be expected. By pure chance, this co-variation could affect the UCM-based analysis. In actual, the authors wrote as follows in the Introduction (page4 line50) “-, along which flexion-extension occurs.”

Experimental design

Page4 line47
In hopping, I think that perturbation to the COM trajectory is more affected by the trunk, head, and upper arm, but was not it necessary to add these segments?

Page8 line134
In hopping, it was written that the authors instructed only “comfortable pace”, but did not the position of the upper limbs or trunk instruct anything? Then it seems that the perturbation to the COM trajectory will change by the effect of trunk and upper limbs.

Validity of the findings

no comment

Additional comments

no comment

---

## Round 0.2 · Minor Revisions

Reviewers agree that with a few minor modifications, the paper is in an acceptable form. Please iron out the last details (i.e., reviewer 1's minor comments) and resubmit.

·

Basic reporting

No comment

Experimental design

No comment

Validity of the findings

No comment

Additional comments

I would like to thank the authors for all the amendments made to their manuscript. The manuscript is now much clearer and informative. Having read through the paper a few times, I only have two very minor comments.

1. Line 164: When describing the coordinate system used in this study, the authors state that the mediolateral (X) axis was pointing 'laterally'. When describing the left limb, this would mean the axis is positive to the left, but when describing the right limb it would mean the axis is positive to the right. I assume this is a mistake, and I am guessing it should read that the axis was positive to the left. Please amend/clarify this in the paper.
2. Line 201: The word 'That' can be uncapitalized here as it is in the middle of a sentence.

Please note that the line numbers refer to those in the PDF document and not the Word document with tracked changes. They are slightly different for some reason.

Reviewer 2 ·

Basic reporting

Thank you for your polite answer. I could understand why you chose a segment angle as an element variable, not a joint angle.

Experimental design

no comment

Validity of the findings

no comment

Additional comments

no comment

---

## Round 0.3 · accepted · Accept

I am happy to see this good piece of work published. It advances of our understanding of motor synergies with a well quantified case of motor abundance, clear methods and meaningful results.